

# RSRNeT: a novel multi-modal network framework for named entity recognition and relation extraction

Min Wang[1,2], Hongbin Chen[1], Dingcai Shen[1,2], Baolei Li[1,2] and Shiyu Hu[1]

[1] School of Mathematics and Computer Science, Gannan Normal University, Ganzhou, China
[2] Key Laboratory of Jiangxi Province for Numerical Simulation and Emulation Techniques, Ganzhou, China

## ABSTRACT

Named entity recognition (NER) and relation extraction (RE) are two important technologies employed in knowledge extraction for constructing knowledge graphs. Uni-modal NER and RE approaches solely rely on text information for knowledge extraction, leading to various limitations, such as suboptimal performance and low efficiency in recognizing polysemous words. With the development of multi-modal learning, multi-modal named entity recognition (MNER) and multi-modal relation extraction (MRE) have been introduced to improve recognition performance. However, existing MNER and MRE methods often encounter reduced efficiency when the text includes unrelated images. To address this problem, we propose a novel multi-modal network framework for named entity recognition and relation extraction called RSRNeT. In RSRNeT, we focus on extracting visual features more fully and designing a multi-scale visual feature extraction module based on ResNeSt network. On the other hand, we also emphasize fusing multi-modal features more comprehensively while minimizing interference from irrelevant images. To address this issue, we propose a multi-modal feature fusing module based on RoBERTa network. These two modules enable us to learn superior visual and textual representations, reducing errors caused by irrelevant images. Our approach has undergone extensive evaluation and comparison with various baseline models on MNER and MRE tasks. Experimental results show that our method achieves state-of-the-art performance in recall and F1 score on three public datasets: Twitter2015, Twitter2017 and MNRE.

# INTRODUCTION

Since the concept of the knowledge graph (KG) proposed by Google in 2012, technologies related to KG have rapidly advanced. Various KGs have emerged, including DBpedia (*Auer et al., 2007*), Wikidata (*Vrandečić & Krötzsch, 2014*), IMGPedia (*Ferrada, Bustos & Hogan, 2017*), and more. A KG is a graphical model utilized to represent and organize knowledge by describing the relationships and attributes between entities, such as people, locations and things. KGs provide significant value in a wide range of real-world applications, such as intelligent question answering (*Yasunaga et al., 2021*), recommendation systems (*Zhou et al., 2020*) and search engines

Corresponding author
Dingcai Shen, dcshensa@gnnu.edu.cn

(*Liu et al., 2020b*). The construction of a KG forms the foundation of KG applications, and the quality of its construction directly impacts the efficiency of these applications.

Named entity recognition (NER) and relation extraction (RE) are two essential technologies used to construct a KG. Traditional uni-modal NER and RE methods primarily depend on text information to build the KG (*Collobert et al., 2011*; *Strubell et al., 2017*; *Huang, Xu & Yu, 2015*). However, these uni-modal methods exhibit several limitations, including poor performance and low efficiency in recognizing polysemous words. With the rapid advancement of multi-modal learning, the integration of multi-modal techniques into NER and RE has gained significant attention. Researchers have developed multi-modal named entity recognition (MNER) and multi-modal relationship extraction (MRE) methods, aiming to leverage image information to enhance the performance of NER and RE tasks. Image information is used as supplementary inputs in MNER and MRE methods to enhance the capabilities of text-based models. Leveraging image information allows these methods to recognize polysemous words and improve the efficiency of entity recognition. Consequently, advancements in MNER and MRE methods (*Moon, Neves & Carvalho, 2018*; *Zheng et al., 2021a*) have demonstrated significant improvements in performance, contributing to the construction of higher-quality knowledge graphs.

The primary challenge in applying the MNER and MRE methods is how to effectively leverage visual information to aid entity recognition and relation extraction while minimizing interference from irrelevant images. *Yu et al. (2020)*; *Zhang et al. (2021a)*; *Zheng et al. (2021a)* demonstrated the effectiveness of incorporating object-level visual features for MNER and MRE tasks. *Xu et al. (2022)* and *Jia et al. (2022)* explored the integration of the entire image features into the textual representations. However, these approaches exhibit low recognition efficiency and struggle with polysemous word recognition. Additionally, RpBERT (*Sun et al., 2021*) proposed training a classifier before the MNER and MRE task to determine the relevance between images and tweets. Nevertheless, this approach heavily relies on large labeled image-text correlation datasets and only focuses on the overall images, disregarding the relevant target objects within the images. Recently, *Chen et al. (2022)* proposed HVPNet. HVPNet introduced a hierarchical visual prefix fusion network designed for MNER and MRE, demonstrating the ability to reduce the negative impact of irrelevant images. However, due to insufficient feature extraction from text and images and an inadequate fusion of multi-modal features, HVPNet still has room for performance improvement.

To address the aforementioned challenges, we propose a novel multi-modal network framework named RSRNeT for MNER and MRE. The network architecture of RSRNeT includes input, multi-scale visual feature extraction, visual feature adaption, multi-modal feature fusion and task output modules. The multi-scale visual feature extraction module utilizes a multi-scale network based on ResNeSt (*Zhang et al., 2022*) to extract visual features from input images. The multi-modal feature fusion module employs RoBERTa (*Liu et al., 2019*) to effectively integrate both visual and textual information. Experimental results show that RSRNeT achieves state-of-the-art (SOTA) performance in recall and F1 metrics on the Twitter2015, Twitter2017, and MNRE benchmark datasets, respectively. The contributions of this article are as follows.

- We propose RSRNeT, a novel end-to-end multi-modal network framework for named entity and relation extraction. RSRNeT utilizes a multi-scale structure to extract visual features more fully and leverages BERT variant to fuse multi-modal features more comprehensively for downstream tasks.
- In RSRNeT, we introduced a multi-granularity visual feature extractor and a multimodal feature fusion module, addressing two key issues in MNER and MRE tasks. The first issue is insufficient image feature extraction. To address this, we devised a multi-scale visual feature extraction method based on ResNeSt network. The second issue is how to extract text features and fuse multi-modal features more comprehensively while minimizing interference from irrelevant images. To address this, we proposed a multi-modal feature fusing method based on variants of BERT.
- RSRNeT achieves SOTA performance. Through evaluation on MNER and MRE tasks, our method achieved SOTA performance on three public datasets. The F1 score in Twitter2015, Twitter2017, and MNRE datasets reached 76.48%, 87.90%, and 83.89%, respectively. Ablation experiments validate the significant roles of each designed module. Furthermore, our method significantly reduces training time. The experimental results demonstrate the effectiveness and superiority of our proposed model.

## RELATED WORK

In this section, we first introduce the representation and fusion of multi-modal features, and pre-trained language models. Then, we offer an overview of related work in both uni-modal and multi-modal methods for NER and RE tasks in KG construction.

### Multi-modal features representation

Multi-modal learning came into existence in the 1970s and were categorized into four eras: the behavioral era, computational era, interaction era and deep learning era (*Baltrusaitis, Ahuja & Morency, 2017*). Multi-modal learning has entered the deep learning era. Multi-modal deep learning leads to a wide range of applications from audio-visual speech recognition, multimedia content indexing and retrieval, understanding human multi-modal behaviors, emotion recognition, multi-modal affect recognition, image and video captioning, visual question answering, multimedia retrieval, to health analysis (*Jabeen et al., 2023*).

In the realm of deep multi-modal learning, five principal technical challenges have surfaced: multi-modal representation, multi-modal translation, multi-modal alignment, multi-modal fusion and multi-modal co-learning (*Baltrusaitis, Ahuja & Morency, 2017*). In this article, we only focus on deep multi-modal representation and fusion. Deep multi-modal representation involves the process of representing information from various media in tensor or vector form. The primary challenge is how to effectively represent and summarize multi-modal data in a manner that leverages complementarity and redundancy (*Jabeen et al., 2023*). The method of multi-modal features representation first learn features from each mode, and then integrates them into a single representation to maximize the retention of common and unique features. Visual features are typically extracted through CNNs or attention mechanisms, while text information is obtained

through text embedding, RNNs, and BERT variants. Subsequently, neural networks models, such as deep Boltzmann machines(DBM) (*Liu et al., 2020a*), auto-encoder (*Khattar et al., 2019*), generative adversarial network (*Zhang et al., 2021b*), and attention mechanisms (*Dai et al., 2021*), are employed to fuse different modal features. The outcome is a unified feature representation tailored for specific applications.

## Multi-modal features fusion and pre-trained language model

Multi-modal feature fusion aims to integrate information from different modes into a unified feature representation. Methods for multi-modal feature fusion can be broadly categorized into three types: simple operations-based, attention-based, and tensor-based approaches. Here, we specifically focus on attention-based approaches. Attention-based fusion methods can be further classified based on the specific objects of the fusing operator. This classification includes image attention, co-attention of images and text, and attention facilitated by transformers. Image attention methods fuse text features and image features obtained through attention mechanisms. For instance, *Zhu et al. (2016)*; *Shih, Singh & Hoiem (2016)* and *Huijuan & Saenko (2016)* explore the incorporation of attention on images using LSTM, RNN, and GRUs models for visual question answering. Co-attention employs symmetric attention structures to generate both attended image feature vectors and attended language vectors (*Lu et al., 2016*). For example, *Gao et al. (2021)* introduces an approach that accelerates the convergence of detection transfomer by incorporating spatially modulated co-attention mechanisms. Regarding attention in transformers, recent studies have emphasized intermediate methods that facilitate fusion to occur on pre-trained language models.

In the context of attention in transformers, recent studies have primarily focused on intermediate methods that enable fusion to occur on pre-trained language models. This fusion process can be categorized into single-stream and dual-stream methods based on their interaction mode (*Su et al., 2019*). In single-stream structures, text and images are combined into a sequence and fed into pre-trained language models, such as BERT, to learn the embedding representation of the context. Representative works in single-stream structures include VL-BERT (*Su et al., 2019*), VisualBERT (*Li et al., 2019*), Unicoder-VL (*Li et al., 2020a*) and UNITER (*Chen et al., 2020*). On the other hand, the dual-stream method encodes different modalities separately, obtaining distinct single-modal features. These single-modal features are then fused at a later stage to create a multi-modal representation. Representative works in dual-stream structures include ViLBERT (*Lu et al., 2019*) and LXMERT (*Tan & Bansal, 2019*). Single-stream methods boast a simpler structure, making them suitable for resource-constrained environments. However, they may encounter challenges when dealing with diverse information and complex tasks. In contrast, dual-stream methods offer improved capability in handling diverse information but come at the cost of increased complexity and resource requirements. In a certain sense, our approach belongs to the dual-stream structure. Our method initially extracts and represents the visual features using a multi-scale extraction and an adaptive module, and then fuses the text and visual features in a variant of BERT. Our approach has led to achieving SOTA performance on three complex datasets.

Multi-modal information fusion is usually implemented using pre-trained language models. In this context, we introduce some pre-trained models that are relevant to our study. Pre-trained language models have revolutionized natural language processing by learning from large amounts of unlabeled data, acquiring general knowledge that significantly enhances performance in downstream tasks. The advantage of utilizing pre-trained models lies in avoiding the need to train a new model from scratch (*Qiu et al., 2020*). RoBERTa (*Liu et al., 2019*) improves upon BERT by implementing several changes. These changes involve adjustments to the masking strategy, eliminating next sentence prediction, optimizing hyperparameters and training on a larger corpus. They lead to better representation learning and higher performance. On the other hand, XLM-RoBERTa (*Conneau et al., 2019*) leverages a large-scale unlabeled dataset from over 100 languages for pre-training. This approach enables the model to acquire a broader range of language knowledge, leading to improved performance on various multilingual tasks. ALBERT (*Lan et al., 2019*) focuses on lightweight model design, with the aim of speeding up training and improving performance.

## Uni-modal NER and RE methods

A knowledge graph is a semantic graph composed of nodes and edges, representing real-world objects and their relationships (*Zhong et al., 2023*). The general process of constructing a knowledge graph includes data acquisition, knowledge extraction, knowledge fusion, and knowledge evolution. NER and RE are two essential technologies for acquiring knowledge in KG construction. NER focuses on identifying entities with specific meanings in the text, such as people, places, things, proper nouns, etc. RE involves extracting factual triples from data and establishing connections between different entities in the text. For the NER and RE tasks, approaches are mainly divided into three categories, *i.e.,* rule-based methods, statistic-based methods and deep learning methods. Rule-based NER methods rely on matching patterns using predefined rules and dictionaries to recognize entities or extract relationships. They often require expert knowledge and manual design of rules.

Previously, statistic-based methods, such as the hidden Markov model (HMM) (*Zhou & Su, 2002*), and the conditional random field (CRF) (*Finkel, Grenager & Manning, 2005*), were used for NER and RE tasks. These statistic-based approaches are based on probability graph model. They treat NER as a sequential classification tagging task, where entities are tagged according to the BIES (beginning, intermediate, ending, single) scheme along with their respective types (*Zhong et al., 2023*). For the RE task, these approaches are popular designs for allowing contextual information to flow through free text and identify relationships between entities. Researchers employ probabilistic-graph (*Mulwad, Finin & Joshi, 2013*), CRF (*Chen & Cafarella, 2013*) and Markov logic network (*Zhu et al., 2009*) to identity relationships.

Recently, deep learning has gained popularity in NER and RE tasks. For the NER task, these models aggregate contextual embeddings according to the input and context encoders, then output word type tags through tag decoders such as a CRF structure or a softmax structure (*Li et al., 2020b*). *Collobert et al. (2011)* were the first to apply CNNs to NER, enabling the extraction of contextual information from text. Following

CNN, IDCNN (*Strubell et al., 2017*) and other variants were proposed for the NER task. Representative uni-modal models of NER include CNN-BiLSTM-CRF (*Ma & Hovy, 2016*), HBiLSTM-CRF (*Lample et al., 2016*) and BERT-CRF. Additionally, PCNN (*Zeng et al., 2015*) and MTB (*Soares et al., 2019*) models are RE-oriented pretraining models based on BERT (*Devlin et al., 2018*). The introduction of Transformer-based models, such as BERT, has revolutionized NER and RE tasks in the context of knowledge graphs. For RE tasks, BiLSTM models (*Zhou et al., 2016*) with inter-word attention can capture long-distance dependencies between entities for relation extraction. Moreover, GNN-based models, such as EPGNN (*Zhao et al., 2019*) and AGGCN (*Guo, Zhang & Lu, 2019*), have shown promising results in relation extraction.

## Multi-modal NER and RE methods

With the increasing prevalence of multi-modal data on social media platforms, MNER and MRE have become important areas of research. MNER aims to improve entity identification by incorporating images as auxiliary inputs alongside text (*Li et al., 2022*). However, the primary challenge in the tasks of MNER and MRE is how to effectively leverage visual information to aid entity recognition and relation extraction while minimizing interference from irrelevant images. Several approaches have been proposed to tackle this challenge. *Zhang et al. (2018)* introduced a co-attention layer that calculates fusion weights between text and image feature vectors. *Moon, Neves & Carvalho (2018)* proposed a multi-modal attention mechanism to selectively extract word-level, character-level, and region-level information from Twitter text, addressing grammatical inconsistencies. *Yu et al. (2020)* pioneered the use of BERT as the text encoder and designed a multi-modal interaction module based on Transformers to model text-image relations, while also addressing bias introduced by irrelevant image features. *Zhang et al. (2021a)* created graphical connections between textual words and visual objects using a visual grounding toolkit. *Yang et al. (2019)* proposed a graph fusion method for graph encoding. *Xu et al. (2022)* introduced a matching and alignment framework to enhance the consistency of representations across different modalities in MNER. Recently, MAF (*Xu et al., 2022*), MRC-MNER (*Jia et al., 2022*) and MNER-QG (*Jia et al., 2023*) use BERT to extract text features and adopt ResNet to extract visual features, respectively. They then perform cross-modal feature matching and fusion before conducting downstream tasks. Specially, *Chen et al. (2022)* proposed HVPNeT, which performed at a SOTA performance in MNER and MRE. However, the inappropriate design of the backbone network results in insufficient feature extraction from text and images and inadequate multi-modal feature fusion. Thus, HVPNet has room for performance improvement.

## METHODOLOGY

Figure 1 illustrates the overall architecture of our proposed multi-modal RSRNeT for MNER and NRE. The RSRNeT consists of five components, including input, multi-scale visual feature extraction, visual feature adaption, multi-modal feature fusion, and task output modules. There are two input modules, *i.e.,* sentence-related images input module and text input module. These two modules are depicted by the gray dashed rectangles in

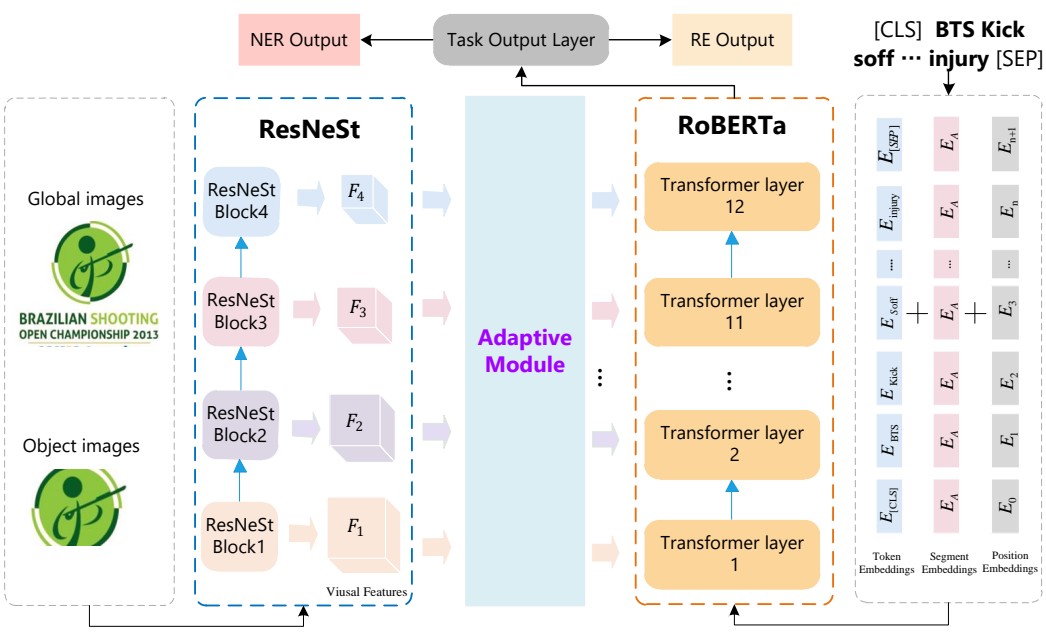

**Figure 1  The overall architecture of RSRNeT.**

Fig. 1. We performed with multi-scale visual feature extraction, utilizing ResNeSt to extract features from the input images. Next, the visual feature adaption step refines the extracted visual features. Subsequently, we employ the RoBERTa to fuse the multi-modal features, effectively integrating both visual and textual information. Finally, the task output modules processed the fused features to execute the MNER and MRE tasks.

## Multi-scale visual feature extraction

The sentence-related images encompass both global images and local target images. A global image refers to the entire image, while a local image represents the region of the target object within a global image. The features extracted from the global image serve to enhance each word representation, while the local image provides the targeted object region as a clue, which is more likely to help us identify some words as the correct entity type (*Zhang et al., 2021a*). By combining both global and local features of respective images, we design a multi-scale visual feature extraction module to capture a more fully visual information. The construction of multi-scale visual feature extraction module is depicted in blue dashed rectangle in Fig. 1. In this module, we use ResNeSt (*Zhang et al., 2022*) as the backbone network to encode global and local images.

ResNeSt, a powerful convolutional neural network model designed for image classification tasks, is depicted in Fig. 2. The structure of ResNeSt incorporates four key components, including split-attention, squeeze-and-excitation (SE) blocks, bottleneck blocks, and feature reuse. Split-attention enhances the model's feature perception by learning the correlations between local features. SE blocks dynamically adjust the importance of features across channels, thereby improving feature representation.

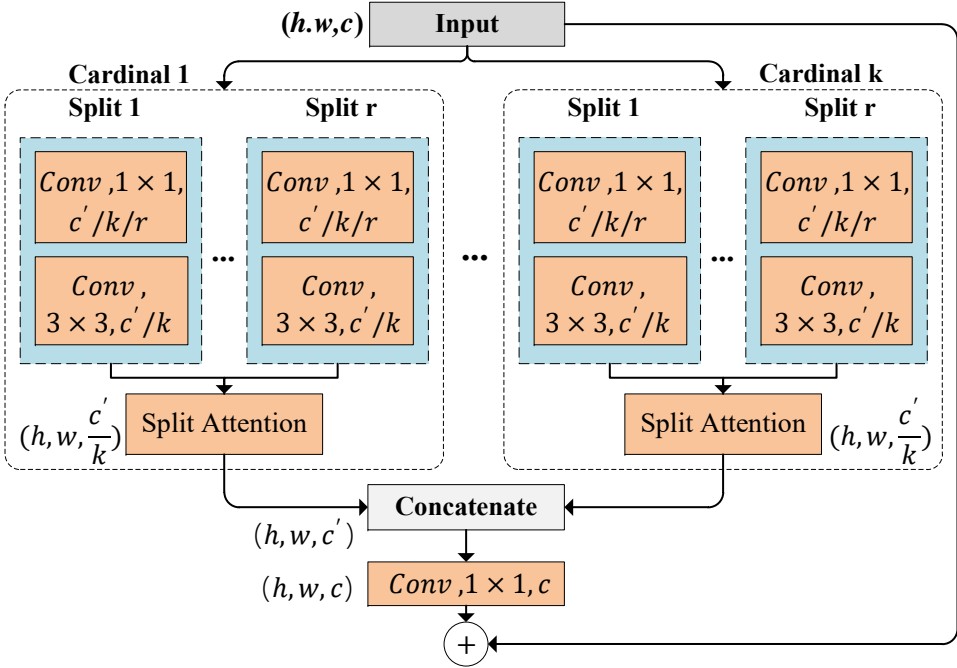

**Figure 2   ResNeSt block.**

Bottleneck blocks reduce computational complexity and parameter count. Feature reuse enables the direct utilization of feature information from preceding blocks. In summary, ResNeSt provides advanced feature representation and perception capabilities, capturing image details and key information effectively. There are two key operations, including feature-map group and split attention in cardinal groups.

**Feature-map group.** The features are divided into several groups, and the number of feature-map groups is determined by a cardinality hyperparameter $K$. We refer to the resulting feature-map groups cardinal groups. A radix hyperparameter $R$ indicates the number of splits within a cardinal group. Thus, the total number of feature groups is $G = KR$. A series of transformations $\{\mathcal{F}_1, \mathcal{F}_2, \ldots, \mathcal{F}_G\}$ is applied to each group, and the representation of each group is $U_i = \mathcal{F}_i(X)$, for $i \in \{1, 2, \ldots, G\}$ .

**Split attention in cardinal groups.** The $k$-th cardinal group is denoted as $\hat{U}^k = \sum_{j=R(k-1)+1}^{RK} U_j$ , where $\hat{U}^k \in \mathbb{R}^{H \times W \times \frac{C}{K}}$ for $k \in 1, 2, \ldots, K$. Here, $H, W$ and $C$ are height, width and channels of the block output, respectively. Global contextual information with embedded channel-wise statistics can be gathered through global average pooling across spatial dimensions $s^k \in \mathbb{R}^{\frac{C}{K}}$. The $c$-$th$ component is calculated by

$$s_c^k = \frac{1}{H \times W} \sum_{i=1}^{H} \sum_{j=1}^{W} \hat{U}_c^k(i,j). \tag{1}$$

The weighted fusion of the cardinal group representation $V^k \in \mathbb{R}^{H \times W \times \frac{C}{K}}$ is aggregated using channel-wise soft attention. In this process, each feature-map channel is obtained

through a weighted combination over splits. The $c$-th channel is calculated as:

$$V_c^k = \sum_{i=1}^{R} a_i^k(c) U_{R(k-1)+i},$$  (2)

where $a_i^k(c)$ denotes a (soft) assignment weight given by:

$$a_i^k(c) = \begin{cases} \dfrac{exp\left(\mathcal{G}_i^c(s^k)\right)}{\sum_{j=0}^{R} exp\left(\mathcal{G}_j^c(s^k)\right)} & if \ R > 1, \\ \dfrac{1}{1 + exp\left(-\mathcal{G}_i^c(s^k)\right)} & if \ R = 1, \end{cases}$$  (3)

Here, mapping $\mathcal{G}_i^c$ determines the weight of each split for the $c$-th channel based on the global context representation $s^k$.

We adopt the visual toolkit introduced by *Zhang et al. (2021a)* to extract the most salient top $m$ local images from each global image. Subsequently, both the global and local images are resized to $224 \times 224$ pixels and used as inputs to the ResNeSt model. We denoted the global image as $\mathcal{I}$ and the visual objects as $\mathcal{O} = \{o_1, o_2, \ldots, o_m\}$. Using ResNeSt, we encode the input image and generate a series of image features at different granularities, denoted as $\{F_1, F_2, F_3, \ldots, F_c\}$. Here, $c = 4$ represents the number of ResNeSt blocks. After the multi-scale visual feature extraction module, we obtained visual features $F_1, F_2, F_3$ and $F_4$ with dimensions size $256 \times 56 \times 56$, $512 \times 28 \times 28$, $1024 \times 14 \times 14$ and $2048 \times 7 \times 7$, respectively. This process enables the extraction of multi-scale image features for subsequent tasks.

## Adaptive module

In this module, we refer to the work of HVPNeT. The adaptive module is to generate vectors, which represent the importance of visual features from the module of multi-scale visual feature extraction. These feature vectors serve as inputs to the multi-modal feature fusion module, which are implemented by a variant framework BERT and will be discussed in the next subsection.

The structure of adaptive module is shown in Fig. 3. It consists of two steps, *i.e.*, project processing and adaptive processing. The projection processing takes multi-scale visual features $\{F_1, F_2, \ldots, F_c\}$ as input, and uses project function $M_\theta(\cdot)$ to transform $F_i$ to features with the same dimension size. The project function $M_\theta(\cdot)$ is expressed by

$$V_c = Conv_{1 \times 1}(F_c),$$  (4)

and

$$V_i = Conv_{1 \times 1}(Pool(F_i)), i = 1, 2, \ldots, c - 1.$$  (5)

where $Pool(\cdot)$ indicates the pooling operation to aggregate the features to the same spatial size. The $Conv_{1 \times 1}$ convolutional layer is used to make the visual features of different granularities match Transformer's embedding size. The Transformer is a layer in multi-modal fusing module, and will be discussed in next subsection. After project processing,

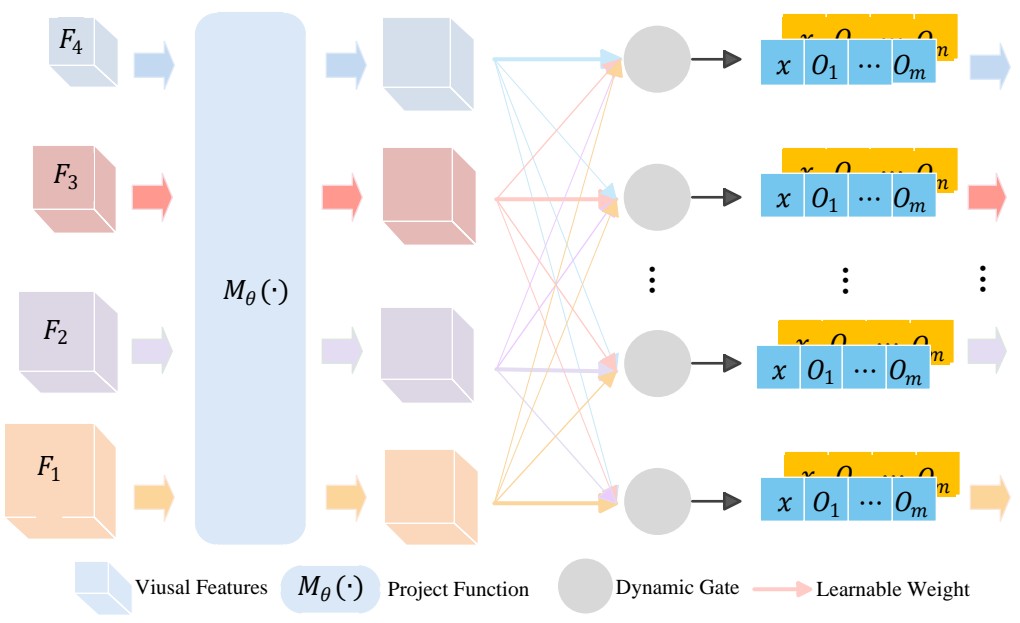

**Figure 3** Adaptive module.

$\{F_1, F_2, \ldots, F_i\}$ are transform to $\{V_1, V_2, \ldots, V_c\}$ features. If $c = 4$, $\{V_1, V_2, V_3, V_4\}$ are with $256 \times 2 \times 2$, $512 \times 2 \times 2$, $1024 \times 2 \times 2$ and $2048 \times 2 \times 2$ dimension size, respectively.

Next, we discuss the adaptive processing. There are four steps to generate visual vectors. Firstly, we used a learnable neural network to obtain a gate signal, which is expressed by

$$\alpha^{(l)} = f\left(W_l\left(\frac{1}{c}\sum_{i=1}^{c} P(V_i)\right)\right), \tag{6}$$

where $P(\cdot)$ is a global average pooling layer, $W_l$ is the $l$th MLP layer, and $f(\cdot)$ is an activation function. It should be noted that our proposed method adopts *SeLU* activation function, whereas the original method uses *Leaky_ReLU*. Thus, we obtained gate signals $\alpha = \{\alpha^l | l \in \{1, 2, \ldots, L\}\}$. Here, $l$ indicates the $l$th Transformer layer, and $L$ is the number of Transform layers in the variant of BERT. Secondly, we calculated the probability vector for the $l$th layer of Transformer by

$$g^{(l)} = Softmax\left(\alpha^{(l)}\right). \tag{7}$$

Thirdly, based on $g^{(l)}$, we derived the final aggregated hierarchical visual features $V_{gated}^{(l)}$, which correspond to the first layer in the $l$th Transformer layer. The $V_{gated}^{(l)}$ is expressed as follows

$$V_{gated}^{(l)} = g^{(l)} V^{(l)}. \tag{8}$$

Since visual information includes feature from global images and local target images, the final visual feature corresponding to the $l$-th Transformer layer was obtained by

$$\widetilde{V}_{gated}^{(l)} = \left[V_{gated}^{(l,I)}; V_{gated}^{(l,o_1)}; \ldots; V_{gated}^{(l,o_m)}\right], \tag{9}$$

where $V_{gated}^{(l,I)}$ is the visual feature of global images, and $V_{gated}^{(l,o_m)}$ are the visual feature of local target images. The $\overset{\sim (l)}{V}_{gated}$ is used to enhance the layer-level representations of the textual modality.

## Multi-modal feature fusion

In this subsection, we address the challenge selecting or improving the BERT model to enhance the adequacy of multi-modal feature fusion for downstream tasks. Specifically, we employed RoBERTa to fuse multi-modal features for MRE tasks, and we utilized XML-RoBERTa to fuse multi-modal features for MNER tasks.

RoBERTa is a pre-trained language model based on the self-attention mechanism, which is improved and optimized based on the BERT model. In comparison to BERT, RoBERTa exhibits two key differences. Firstly, RoBERTa eliminates the next sentence prediction task present in BERT, retaining only the masked language modeling task. This modification allows it to concentrate more on learning contextual information and enhancing language representation capabilities. Secondly, RoBERTa employs the byte-pair encoding (BPE) tokenization algorithm, whereas BERT uses the wordpiece tokenization algorithm. The BPE tokenization algorithm splits the text into subwords, offering better handling of unknown vocabulary and multilingual scenarios. Similarly, XML-RoBERTa also is a pre-trained language model specifically designed for handling multilingual text. It leverages a large-scale unlabeled dataset from over 100 languages for pre-training, enabling the model to acquire a broader range of language knowledge.

The multi-modal feature fusion processes are similar with the RoBERTa and XML-RoBERTa models. Here, we take the RoBERTa model as an example to present the steps of multi-modal feature fusion. As shown in Fig. 1, the multi-modal feature fusing module has $L = 12$ Transformer layers, each represented in Fig. 4. For the $l$-th Transformer layer, the $l$-th visual gated feature $\overset{\sim (l)}{V}_{gated}$ and the output of $(l-1)$-th Transformer, denoted by $H^{(l-1)}$, are used as input. It is should be noted that the first Transformer layer takes textual sequence $X$ as $H^{(0)}$. We integrate the multi-scale visual feature information as a prefix of textual information at each self-attention layer in the RoBERTa model. Specifically, for $l$-th Transformer layer, given an input sequence $X = \{x_1, x_2, \ldots, x_n\}$, the contextual representation $H^{l-1} \in \mathbb{R}^{n \times d}$ is first projected into the query/key/value vector, respectively:

$$Q^l = H^{l-1} W_l^Q, K^l = H^{l-1} W_l^K, V^l = H^{l-1} W_l^V, \tag{10}$$

where $W_l^Q$, $W_l^K$, and $W_l^V$ are learnable projection matrices. These matrices have the same dimensions as $H^{l-1}$. For the aggregated hierarchical visual features $\overset{\sim (l)}{V}_{gated}$, in the self-attention module of $l$-th layer, we used a set of linear transformations $W_l^\varnothing \in \mathbb{R}^{d \times 2 \times d}$ to project them into the same embedding space of textual representation. Next, we defined the visual prompt $\varnothing_k^l, \varnothing_v^l \in \mathbb{R}^{hw(m+1) \times d}$ operation as:

$$\{\varnothing_k^l, \varnothing_v^l\} = \overset{\sim (l)}{V}_{gated} W_l^\varnothing. \tag{11}$$

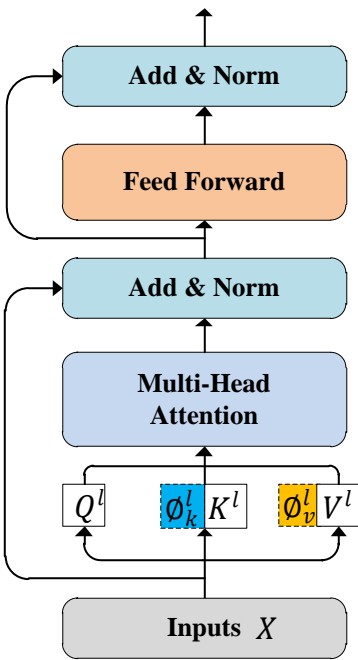

**Figure 4  Transformer layer.**

Here, $hw(m+1)$ denotes the length of the visual sequence, and $m$ denotes the number of visual objects detected by the target detection algorithm. Formally, the attention based on visual prefixes was calculated as follows:

$$Prefix\_Attention^l = softmax\left(\frac{Q^l\left[\varnothing_k^l;K^l\right]^T}{\sqrt{d}}\right)\left[\varnothing_v^l;V^l\right], \tag{12}$$

where $\left[\varnothing_k^l;K^l\right]$ denotes the concatenation of $\varnothing_k^l$ and $K^l$, $\left[\varnothing_v^l;V^l\right]$ denotes the concatenation of $\varnothing_v^l$ and $V^l$.

## Classifier

Through the variant of BERT, we obtained the final representation of multi-modal feature, $H^L = U\left(X, \overset{\sim}{V}_{gated}^{(l)}\right)$. Here, $U(\cdot)$ denotes the visual prefix-based attention operation. Finally, we used different classifier layers for the NER and RE tasks, respectively.

**Named entity recognition.** We used the CRF decoder for the NER task, which is similar to the method proposed by *Moon, Neves & Carvalho (2018)*. We provide the final hidden vector of BERT $H^L$ to the CRF model. For a sequence of labels $\mathbf{y} = \{y_1, \ldots, y_n\}$ , the probability of $y$ and the objective of NER are defined as follows:

$$p\left(y|H^L\right) = \frac{\prod_{i=1}^n S_i\left(y_{i-1}, y_i, H^L\right)}{\sum_{y' \in Y}\prod_{i=1}^n S_i\left(y_{i-1}', y_i', H^L\right)}, \tag{13}$$

**Table 1  The details of MNER datasets.**

| Dataset | Train | Dev | Test | Avg length |
|---|---|---|---|---|
| Twitter-2015 | 4,000 | 1,000 | 3,257 | 95 |
| Twitter-2017 | 4,290 | 1,432 | 1,459 | 64 |

**Table 2  The details of the MNRE dataset.**

| Dataset | Sentence | Entity | Relation | Image |
|---|---|---|---|---|
| MNRE | 9,201 | 30,970 | 23 | 9,201 |

$$\mathcal{L}_{ner} = -\sum_{i=1}^{M} log\left(p\left(y^{(i)}|U\left(X^{(i)}, \widetilde{V}_{gated}\right)\right)\right). \tag{14}$$

where $Y$ denotes the predefined set of tags with BIO (B-begin, I-inside, O-outside) tagging schema, and $S(\cdot)$ denotes the potential functions (*Lample et al., 2016*).

**Relation extraction** Assume an RE dataset represent as $\mathcal{D}_{re} = \{(X^{(i)}, r^{(i)})\}_{i=1}^{M}$, the objective of RE is to predict the relation $r \in \mathcal{Y}$ between subject entities and object entities. A [CLS] head is employed to compute the probability distribution over the class set $\mathcal{Y}$ using the softmax function $p(r|X) = \text{Softmax}([\mathbf{W}H]_{[CLS]}\mathbf{L})$. Refer to (*Chen et al., 2022*), the parameters of $\mathbf{L}$ and $\mathbf{W}$ are fine-tuned by minimizing the cross-entropy loss function $\mathcal{L}_{re}$ over $p(r|X)$ on the entire $\mathcal{X}$.

$$\mathcal{L}_{re} = -\sum_{i=1}^{M} log\left(p\left(r^{(i)}|U\left(X^{(i)}, \widetilde{V}_{gated}\right)\right)\right). \tag{15}$$

# EXPERIMENTS AND RESULTS

In this section, we thoroughly evaluate our proposed approach through experiments conducted on MNER and MRE tasks. For visual feature extraction, we utilize ResNeSt50 (*He et al., 2016*) as our backbone network. As for text encoding, we employ RoBERTa-base (*Liu et al., 2019*) and XLM-RoBERTa (*Conneau et al., 2019*). The results demonstrate the superiority of our proposed approach over other unimodal and multi-modal methods on the same publicly available datasets.

## Datasets
We utilized Twitter-2015 (*Zhang et al., 2018*) and Twitter-2017 (*Lu et al., 2018*) datasets for the task of MNER, and MNRE (*Zheng et al., 2021b*) datesets for the task of MRE. The details of Twitter-2015 and Twitter-2017 can be found in Table 1. And Table 2 provides the details of the MNRE dataset.

## Experimental setup
The proposed approach was implemented using PyTorch 1.8.1. The training and testing of the network models were conducted on a computer server, which equipped with an

Intel Xeon Gold 4260 CPU, 4 T V100 GPUs, 96 GB of DDR4 memory and CentOS 7.9 operating system. For optimization, the AdamW (*Loshchilov & Hutter, 2017*) optimizer was employed. The learning rate was gradually increased to a maximum value during the first 10% of the gradient update using linear warm-up, and then it was linearly decayed for the remainder of the training. The weight decay for all non-biased parameters was set to 0.01. Additionally, the number of image objects used was set to 3.

For the MNER task, we set the batch size to 16, and conducted experiments with different learning rate intervals [1e−5, 3e−5]. The training process was carried out over 30 epochs, with validation performed after the 16th epoch. Regarding the MRE task, we employed a batch size of 16 with a fixed learning rate 1e−5. The model was trained for 15 epochs, and validated after the 8 *th* epoch. In both tasks, we selected the model that exhibited the best performance on the validation set and the test set.

## Baseline models

To demonstrate the superiority of our approach, we conducted a comprehensive comparison between our model and several baseline models . This comparison involved evaluating our model against three groups of models: text-based models, the previous SOTA models of MNER and MRE, and other features fusion models.

- Text-based models: We first consider a representative set of text-based models, including CNN-BiLSTM-CRF (*Ma & Hovy, 2016*), HBiLSTM-CRF (*Lample et al., 2016*) and BERT-CRF for NER. PCNN (*Zeng et al., 2015*) and MTB (*Soares et al., 2019*) models are for RE task. MTB is an RE-oriented pretraining model based on BERT.
- Previous SOTA models: We also considered another set of SOTA multi-modal methods previously used for MNER and MRE tasks. For the MNER task, we examined the latest SOTA models, including AdapCoAtt-BERT-CRF (*Zhang et al., 2018*), OCSGA (*Wu et al., 2020*), UMT (*Yu et al., 2020*), and UMGF (*Zhang et al., 2021a*). UMGF proposes a unified multi-modal graph fusion approach. For the MRE task, we explored the latest SOTA models, such as BERT+SG (*Soares et al., 2019*), and MEGA (*Zheng et al., 2021a*). BERT+SG connects the textual representation of BERT with visual features generated by the scene graph (SG) tool (*Tang et al., 2020*), while MEGA proposes a dual graph for multi-modal alignment to capture correlations between entities and objects for better performance. Additionally, we considered two other SOTA models, VisualBERT (*Li et al., 2019*) and HVPNet (*Chen et al., 2022*). Unlike the aforementioned SOTA approaches which primarily rely on co-attention, VisualBERT is a single-stream structure, serving as a strong baseline for comparison.
- Other features fusion models: Further, we conducted comparisons with recent methods such as MAF (*Xu et al., 2022*), MRC-MNER (*Jia et al., 2022*) and MNER-QG (*Jia et al., 2023*). The three approaches utilize BERT for extracting text features and ResNet for extracting visual features. Subsequently, they employ cross-modal feature matching and fusion before proceeding to downstream tasks. In comparison to MAF, MRC-MNER and MNER-QG methods incorporate queries from machine reading comprehension to retrieve relevant visual information in linguistic contexts. It is worth noting that,

**Table 3** Performance comparison between RSRNet and baseline method (The best performance is indicated in bold).

| Modality | Methods | Twitter-2015 | | | Twitter-2017 | | | MNRE | | |
|---|---|---|---|---|---|---|---|---|---|---|
| | | Precision | Recall | F1 | Precision | Recall | F1 | Precision | Recall | F1 |
| Text | CNN-BiLSTM-CRF | 66.24 | 68.09 | 67.15 | 80.00 | 78.76 | 79.37 | – | – | – |
| | HBiLSTM-CRF | 70.32 | 68.05 | 69.17 | 82.69 | 78.16 | 80.37 | – | – | – |
| | BERT-CRF | 69.22 | 74.59 | 71.81 | 83.32 | 83.57 | 83.44 | – | – | – |
| | PCNN | – | – | – | – | – | – | 62.85 | 49.69 | 55.49 |
| | MTB | – | – | – | – | – | – | 64.46 | 57.81 | 60.86 |
| Text+Image | AdapCoAtt-BERT-CRF | 69.87 | 74.59 | 72.15 | 85.13 | 83.2 | 84.1 | – | – | – |
| | OCSGA | 74.71 | 71.21 | 72.92 | – | – | – | – | – | – |
| | UMT | 71.67 | 75.23 | 73.41 | 85.28 | 85.34 | 85.31 | 62.93 | 63.88 | 63.46 |
| | UMGF | 74.49 | 75.21 | 74.85 | 86.54 | 84.5 | 85.51 | 64.38 | 66.23 | 65.29 |
| | BERT+SG | – | – | – | – | – | – | 62.95 | 62.65 | 62.80 |
| | MEGA | 70.35 | 74.58 | 72.35 | 84.03 | 84.75 | 84.39 | 64.51 | 68.44 | 66.41 |
| | VisualBERT | 68.84 | 71.39 | 70.09 | 84.06 | 85.39 | 84.72 | 57.15 | 59.48 | 58.3 |
| | HVPNeT | 73.87 | 76.82 | 75.32 | 85.84 | 87.93 | 86.87 | 83.64 | 80.78 | 81.85 |
| | MAF | 71.86 | 75.10 | 73.42 | 86.13 | 86.38 | 86.25 | – | – | – |
| | MRC-MNER | **78.10** | 71.45 | 74.63 | **88.78** | 85.00 | 86.85 | – | – | – |
| | MNER-QG | 77.76 | 72.31 | 74.94 | 88.57 | 85.96 | 87.25 | – | – | – |
| | RSRNeT-woRoB | – | – | – | – | – | – | 84.21 | 82.50 | 82.85 |
| | RSRNeT-woXlm | 75.47 | 76.91 | 75.89 | 86.91 | 87.49 | 87.22 | – | – | – |
| | RSRNeT-woRSt | 75.22 | 76.53 | 75.37 | 86.21 | 86.95 | 86.75 | 83.82 | 82.01 | 82.21 |
| | RSRNeT-plain | 75.01 | 76.18 | 75.12 | 86.06 | 86.78 | 86.33 | 83.27 | 81.79 | 82.02 |
| | RSRNeT-1T3 | 75.13 | 76.35 | 75.19 | 86.12 | 86.86 | 86.50 | 86.50 | 81.90 | 82.17 |
| | **RSRNeT** | 75.83 | **77.35** | **76.48** | 87.55 | **88.21** | **87.90** | **84.78** | **83.06** | **83.89** |

among these models, MNER-QG currently represents the SOTA model for MNER, while HVPNeT holds the position of the latest SOTA model for MRE.

## Experimental results

In this subsection, we conduct a comprehensive comparison of our proposed method with various baseline approaches. Firstly, we present the performance comparison between RSRNeT and baseline models. Secondly, we present the results of our ablation study. Thirdly, we compare the training time of our approach with that of other methods. Fourthly, we present the results obtained under the low-resource scenario. Finally, we provide the comparison results of case study.

### *Performance comparison*

The performance comparison between RSRNeT and baseline models on the three datasets are presented in Table 3. Several key observations can be drawn from the experimental results. Firstly, compared to text-based methods, it is evident that multi-modal approaches achieve better performance in both NER and RE tasks. For example, in NER task, the F1 score of UMGF improved 2.0%, compared to BERT-CRF. Similarly, MEGA improved by 5.55% in F1 score for RE task, compared to MTB.

Secondly, our proposed method outperforms the SOTA model HVPNeT, achieving improvements of 1.16%, 1.03%, and 2.04% in F1 scores on the Twitter-2015, Twitter-2017 and MNRE datasets, respectively. Our method and HVPNeT all utilized multi-scale visual prefixes as textual cues to reduce the errors and interference caused by irrelevant images. From the results of performance comparison, our method extracted and fused multi-modal features more effectively, resulting in improving performance in MNER and MRE tasks.

Thirdly, the comparison results with other fusion methods are presented in Table 3. Analyzing the data, we observe that RSRNet achieved the best performance in F1 and recall metrics on Twitter-2015 and Twitter-2015 dataset. Specially, our proposed method outperformed MNER-QG, which is the best approach among other fusion methods, achieving improvements of 1.54% and 0.65% in F1 scores on the Twitter-2015 and Twitter-2017 datasets, respectively. However, it's worth noting that the precision of our method lags behind that of MRC-MNER and MNER-QG. The reason is that MRC-MNER and MNER-QG leverage queries in machine reading comprehension, potentially enabling more accurate localization of visual regions and improved modeling of cross-/within-modal relations.

### Ablation study

To validate the effectiveness of our design, we conducted ablation experiments, ensuring a fair comparison by setting identical parameters for each variant model. The results of these experiments are presented at the last six rows in Table 3.

RSRNeT-woRoB: In this ablation experiment, we substituted the RoBERTa-base module with BERT-base to assess the efficacy of RoBERTa-base for MRE. Since RoBERTa is only used in the MRE task, this experiment focuses on MRE task. The results show a 1.04% decrease in the F1 score on the MNRE dataset compared to RSRNeT. This decline indicates the importance of the RoBERTa-base module for MRE and highlights how it improves the effectiveness of text feature extraction.

RSRNeT-woXlm: This ablation experiment involves removing the XLM-RoBERTa module and replacing it with BERT-base to assess the effectiveness of XLM-RoBERTa. Since XLM-RoBERTa is only used in the MNER task, this experiment focused on MNER. Compared to RSRNeT, RSRNeT-woXlm has 0.57% and 0.67% decrease in F1 score on the Twitter-2015 and Twitter-2017 datasets, respectively. These results indicate that the XLM-RoBERTa module enhances the efficiency of text feature extraction, proving to be beneficial for MNER task. This ablation experiment provides confirmation of the effectiveness and significance of our method.

RSRNeT-woRSt: In this ablation experiment, RSRNeT replaces the ResNeSt-50 module with ResNet-50, aims to evaluate the effectiveness of ResNeSt-50. The results of this experiment show a decrease of 1.11%, 1.14%, and 1.68% in F1 score on the Twitter-2015, Twitter-2017, and MNRE datasets, respectively, compare to RSRNeT. These results demonstrate that the ResNeSt-50 module enhances the efficiency of images feature extraction and improves the utilization of images information. Therefore, this ablation experiment provides empirical evidence to support the effectiveness and significance of our method utilizing ResNeSt-50.

**Table 4   Training time comparison between RSRNet and HVPNet.**

| Methods | Twitter-2015 | Twitter-2017 | MNRE | Avg time(s) |
|---|---|---|---|---|
| HVPNeT | 5,534 | 5,468 | 7,475 | 6,159 |
| RSRNeT | 2,890 | 2,832 | 7,065 | 4,262 |

RSRNeT-plain: This variant of RSRNeT omits the multi-scale visual feature extractor. Specifically, visual features are assigned using the output of the 4th block of ResNeSt. Subsequently, an adaptive module is employed to map these visual features to each layer corresponding to RoBERTa. Compare to RSRNeT, the results reveal a decrease of 1.36%, 1.57%, and 1.87% in F1 score on the Twitter-2015, Twitter-2017, and MNRE datasets, respectively. This demonstrates that multi-scale visual feature extractor is indispensable, facilitating more comprehensive feature extraction.

RSRNeT-1T3: To explore the impact of adaptive module in RSRNeT, we designed a simplified adaptive processing. Specifically, given that ResNeSt consists of four blocks and RoBERTa comprises twelve layers, we intuitively design a method to directly map projection visual features from one block to the three layers in RoBERTa, *i.e.,* map $V_i$ to $\overset{\sim}{V}_{gated}^{(3i-2)}$, $\overset{\sim}{V}_{gated}^{(3i-1)}$ and $\overset{\sim}{V}_{gated}^{(3i)}$. We refer to this variant as RSRNeT-1T3. Compare to RSRNeT, the results of this experiment indicate a decrease of 1.29%, 1.4%, and 1.72% in F1 score on the Twitter-2015, Twitter-2017 and MNRE datasets, respectively. This demonstrates that adaptive module can achieve better performance.

### Training time comparison

In comparison with HVPNet, we conducted an evaluation of the training time for both the NER and RE tasks. The comparison results are presented in Table 4. The experimental result reveals that our method significantly reduces the training time for the NER task. Additionally, our approach also exhibits a slight reduction in training time for the RE task. On average, our method achieves substantially lower training time than HVPNet. Upon closer examination, we attribute the reduction in training time primarily to the extended training duration and larger batch size employed by RoBERTa and XLM-RoBERTa, compared to BERT.

### Low-resource scenario

To evaluate the performance of our method in a low-resource setting, we conducted experiments by randomly sampling 5% to 50% of the data from the original training set, thereby creating low-resource training sets. The results of these experiments are presented in Fig. 5, offering a performance comparison between our method and several baseline methods in the low-resource scenario. The analysis of these results yields the following observations: firstly, in the low-resource scenario, our method, HVPNet, UMT, and MEGA consistently outperformed BERT-CRF. This demonstrates that incorporating multi-scale visual features remains beneficial for NER and RE tasks even with limited resources. Secondly, from the Fig. 5, RSRNeT exhibits superior performance compared to other baselines, further confirming the effectiveness and data efficiency of our proposed method.

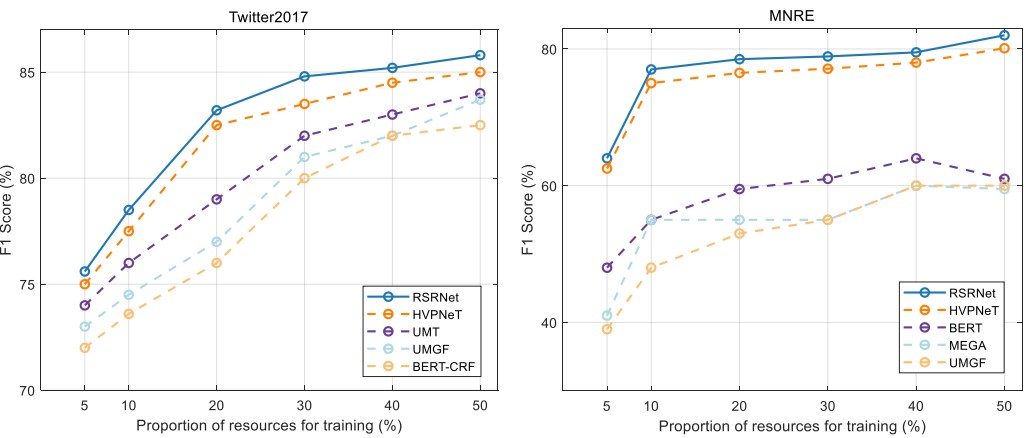

**Figure 5** Performances comparison on low-resource scenario.

## Case study

To validate the effectiveness and robustness of our method, we conducted case studies, as shown in Table 5. We present results from two groups of experiments. For each group, we selected different image-text pairs with varying relevance from the MNRE dataset (*Zheng et al., 2021b*), as shown in the first two rows in a group. The subsequent rows indicate the entity types of the selected text and the relationship labels between entities. The last rows display the predicted entity types and relationships between entities for these test samples. We conduct this experiment using various methods, including BERT (*Devlin et al., 2018*), VisualBERT (*Li et al., 2019*), MEGA (*Zheng et al., 2021a*), HVPNeT (*Chen et al., 2022*) and our proposed RSRNet.

From the Table 5, it can be observed that in scenarios with high semantic correlation between image and text, most methods successfully identified the entity types and relationships, which between entities in image-text pairs. In scenarios with weak semantic correlation between image and text, only our method RSRNeT and a few others can successfully capture more features and accurately predict entity types and relationships. Specially, in scenarios with irrelevant image-text pair, our method RSRNeT continues to predict all cases correctly, whereas the other four methods almost failed. These experimental results clearly demonstrate that our method performs well in multiple case studies, indicating that our model effectively alleviates the negative impact of irrelevant image pairs on entity and relationship recognition. This showcases the robustness and effectiveness of our approach.

## CONCLUSIONS

In this article, we proposed a novel multi-modal named entity and relationship extraction network model named RSRNeT. Our approach utilizes hierarchical and multi-scale images information processed by the ResNeSt network, serving as prefixes for each self-attention layer in RoBERTa. This unique design allows us to effectively incorporate image feature representation into the NER and RE tasks, while mitigating the impact of unrelated images

**Table 5   Case analysis.**

| Relevant image-text pair | Weak relevant image-text pair | Irrelevant image-text pair |
|---|---|---|
| The winter scenery of Jinshanling Great Wall Hebei Province of China | Baby female Dachshund Beagle mix in Redondo Beach CA | Bill Cosby's Hollywood Walk of Fame vandalized |
| 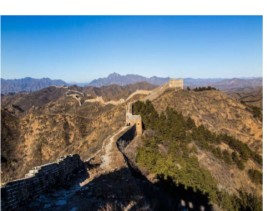 | 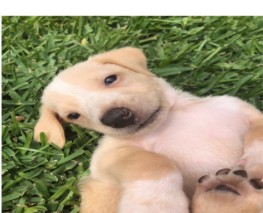 | 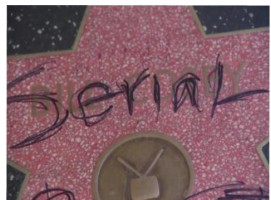 |
| **Gold Relations: loc/loc/contain** | **/loc/loc/contain** | **per/misc/present_in** |
| BERT:          loc/loc/contain ✓ | per/per/present_in  ✗ | misc/org/present_in ✗ |
| VisualBERT: loc/per/contain ✗ | loc/loc/contain        ✓ | per/loc /present_in  ✗ |
| MEGA:        loc/loc/contain ✓ | misc/per/member_of ✗ | loc/loc/contain        ✗ |
| HVPNeT:     loc/loc/contain ✓ | per/org/member_of  ✗ | loc/loc/present_in   ✗ |
| **RSRNeT:**     loc/loc/contain ✓ | loc/loc/contain        ✓ | per/misc/present_in ✓ |
| Best Day Trips from Zagreb Tour Croatia | Commercial in US Michigan Belleville MondProp | Come and see Billo at Clerkenwell Design Week |
| 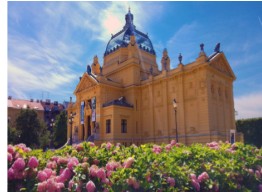 | 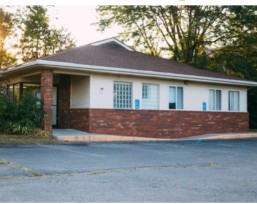 | 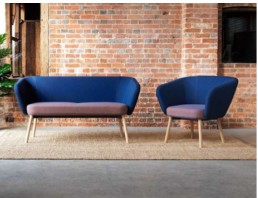 |
| **Gold Relations:/loc/loc/contain** | **/loc/loc/contain** | **/per/misc/present_in** |
| BERT:          loc/per/member_of ✗ | loc/per/present_in ✗ | per/org/member_of ✗ |
| VisualBERT: loc/loc/contain ✓ | per/per/peer        ✗ | org/misc /contain    ✗ |
| MEGA:        loc/loc/contain ✓ | loc/per/present_in ✗ | loc/per/contain      ✗ |
| HVPNeT:     loc/loc/contain ✓ | loc/loc/contain     ✓ | per/misc/present_in ✓ |
| **RSRNeT:**     loc/loc/contain ✓ | loc/loc/contain     ✓ | per/misc/present_in ✓ |

and enhancing recognition accuracy. Extensive experiments conducted on three public datasets, experiment results show that RSRNeT achives the SOTA performance on recall and F1 score, compared to other baseline methods.

In the future, we plan to explore the image-text matching capabilities of multimodal pre-trained models to assess the relevance of image-text pairs, thereby mitigating the adverse effects of irrelevant images on MNER performance. Additionally, our future research will focus on the construction and application of knowledge graphs using large language models.

## ACKNOWLEDGEMENTS

We thank the anonymous reviewers for their insightful comments.

## Funding

This research was funded by the National Natural Science Foundation of China (No.62362002), the Nature and Science Foundation of Jiangxi Province of China (Nos. 20212BAB202003, 20224BAB212022), the Science and Technology Project of Education Bureau of Jiangxi province (Nos. GJJ201401, GJJ211435, GJJ211405), and the open project funding of Key Laboratory of Jiangxi Province for Numerical Simulation and Emulation Techniques, China (No. NSET20201201). The funders had no role in study design, data collection and analysis, decision to publish, or preparation of the manuscript.

## Grant Disclosures

The following grant information was disclosed by the authors:
National Natural Science Foundation of China: 62362002.
Nature and Science Foundation of Jiangxi Province of China: 20212BAB202003, 20224BAB212022.
Science and Technology Project of Education Bureau of Jiangxi province: GJJ201401, GJJ211435, GJJ211405.
The open project funding of Key Laboratory of Jiangxi Province for Numerical Simulation and Emulation Techniques, China: NSET20201201.

## Competing Interests

The authors declare there are no competing interests.

## Author Contributions

- Min Wang conceived and designed the experiments, analyzed the data, performed the computation work, prepared figures and/or tables, authored or reviewed drafts of the article, and approved the final draft.
- Hongbin Chen conceived and designed the experiments, performed the experiments, performed the computation work, authored or reviewed drafts of the article, and approved the final draft.
- Dingcai Shen analyzed the data, prepared figures and/or tables, authored or reviewed drafts of the article, and approved the final draft.
- Baolei Li performed the experiments, performed the computation work, prepared figures and/or tables, authored or reviewed drafts of the article, and approved the final draft.
- Shiyu Hu analyzed the data, authored or reviewed drafts of the article, and approved the final draft.

## Data Availability

The code is available at GitHub and Zenodo:
- https://github.com/binbinbin666/RSRNeT/tree/main/RSRNeT.
- chen, hongbin, & wang, min (2023). This is the code for the article: RSRNeT: A novel multi-modal network framework for named entity recognition and relation extraction. Zenodo. https://doi.org/10.5281/zenodo.10148773

The Twitter 2015 and 2017 datasets are available at figshare:

chen, hongbin (2023). This is the dataset for the article: RSRNeT: A Novel Multi-modal Network Framework for Named Entity Recognition and Relation Extraction. figshare. Dataset. https://doi.org/10.6084/m9.figshare.24585732.v1.

MNRE is available in at GitHub: https://github.com/thecharm/MNRE.

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
