# Peer review of "RSRNeT: a novel multi-modal network framework for named entity recognition and relation extraction"

_PeerJ Computer Science, doi:10.7717/peerj-cs.1856_

## Round 0.1 · original submission · Major Revisions

Both reviewer agree the paper's contributions are relevant, yet the paper need revisions.

I would like to point out three requests:

- revision of the related work section to better clarify the background, scope and impact of the proposed method;

- an overall check of grammar and improvement of the definition of the main concepts presented in the paper;

- additional experiments to compare with more recent models and to better explore the contribution of the various components of the proposed method.

**Language Note:** The review process has identified that the English language must be improved. PeerJ can provide language editing services - please contact us at copyediting@peerj.com for pricing (be sure to provide your manuscript number and title). Alternatively, you should make your own arrangements to improve the language quality and provide details in your response letter. – PeerJ Staff

Reviewer 1 ·

Basic reporting

a. In this paper, there are some grammatical errors that should be carefully reviewed and corrected by the authors.

b. The introduction clearly provides the background of multi-model NER and RE.

c. The structure of this paper is well-organized.

d. The access to the data used in the experiments is provided.

e.There are missing explanations for some concepts, such as "local image" and "global image".

Experimental design

a. The authors utilize an adaptive module to transform visual features into Roberta's embedding space. Their methods demonstrate promising performance on 3 multi-modal datasets.

b. While the authors use ResNeSt and Roberta as the visual and text modules respectively, they neglect recent advancements in large-scale foundation models. Comparing them with recent visual models (CLIP, BLIP, llava) and language models (llama2, flan-t5) would provide a deeper understanding of these tasks.

c. The ablation study primarily focuses on the effect of different base models but fails to demonstrate the effect of the framework structure. It would be inspiring to compare the prefix attention multi-modal feature fusion method with other feature fusion methods.

Validity of the findings

a. The authors claim that their method is more robust towards irrelevant image-text pairs, but they only provide one case study without quantitative analysis. Additionally, they do not provide insight into why this framework can be more robust.

b. This paper presents detailed experimental settings, making it easier to reproduce.

Additional comments

a. This paper shows promising results on 3 public datasets under multi-modal IE tasks. It also provides access to the data and details of the experiment settings.

b. But the grammatical errors and missing explanation for concepts impair the readability.

c. Further experiments should be conducted to explore the impact of the framework structure and keep abreast of the latest advancements in the multi-modal field.

Reviewer 2 ·

Basic reporting

Dear author,
This manuscript proposed a novel multi-modal network framework for named entity and relation extraction, and design a multi-scale visual feature extractor and a multi-modal feature fusing method. But there are also some shortcomings in this manuscript. The specific comments are as follows:
1.In the section of RELATED WORK, this manuscript introduced the single-stream and dual-stream structures, what are the advantages and disadvantages of each structure? Which structure is used in this manuscript and why?
2.This manuscript is mainly aimed at multi-modal NER and RE tasks, so in the RELATED WORK, it would be better to focus on the introduction of multi-modal feature processing technologies and problems. This may better highlight the contributions of this manuscript.

Experimental design

1.Does “y” in line 288 mean the same as “y” in line 294? Because it looks like one represents the label sets of entity and the other represents the label sets of relationship.
2.In Figure 2, it is "Split" instead of "Sprit".

Validity of the findings

1.The structure of this manuscript is clear, but it is less innovative. ResNeSt and HVPNeT are both models that have been proposed.

---

## Round 0.2 · Minor Revisions

The paper sensibly improved from the previous version. As a reviewer pointed out, there are minor edits required to consider it acceptable for publication. Please make the requested corrections.

Reviewer 2 ·

Basic reporting

Dear author,
This manuscript proposed a novel multi-modal network framework for named entity recognition and relation extraction called RSRNeT. It's great to see this manuscript again. Now, the author has basically solved our concerns, and the quality of the manuscript has also been greatly improved. Thanks to the authors for their work. But there are also some shortcomings in this manuscript. The specific comments are as follows:
1.In Figure 1, should it be "Task output Layer" instead of "Task outpput Layer"?
2.It would be better to add to Figure 1 the appropriate interpretation of variables such as F1, EA, and E0.
3.In the REFERENCES section, it is better to appropriately increase the proportion of articles in the last three years.

Experimental design

no comment

Validity of the findings

no comment

Annotated reviews are not available for download in order to protect the identity of reviewers who chose to remain anonymous.

---

## Round 0.3 · accepted · Accept

I have reviewed this minor revision of the paper. The requests from the previous revision round have been all addressed. The paper is now ready for publication, congratulations.